# Identification of miRNAs Involving Potato-*Phytophthora infestans* Interaction

**DOI:** 10.3390/plants12030461

**Published:** 2023-01-19

**Authors:** Ming Luo, Xinyuan Sun, Meng Xu, Zhendong Tian

**Affiliations:** 1National Key Laboratory for Germplasm Innovation & Utilization of Horticultural Crops, Huazhong Agricultural University (HZAU), Wuhan 430070, China; 2Key Laboratory of Potato Biology and Biotechnology (HZAU), Ministry of Agriculture and Rural Affairs, Wuhan 430070, China; 3Potato Engineering and Technology Research Center of Hubei Province (HZAU), Wuhan 430070, China; 4Hubei Hongshan Laboratory (HZAU), Wuhan 430070, China

**Keywords:** potato, *Phytophthora infestans*, miRNA, disease resistance, transboundary regulation

## Abstract

sRNAs (small RNAs) play an important role in regulation of plant immunity against a variety of pathogens. In this study, sRNA sequencing analysis was performed to identify miRNAs (microRNAs) during the interaction of potato and *Phytophthora infestans*. Totally, 171 potato miRNAs were identified, 43 of which were annotated in the miRNA database and 128 were assigned as novel miRNAs in this study. Those potato miRNAs may target 878 potato genes and half of them encode resistance proteins. Fifty-three potato miRNAs may target 194 *P. infestans* genes. Three potato miRNAs (novel 72, 133, and 140) were predicted to have targets only in the *P. infestans* genome. miRNAs transient expression and *P. infestans* inoculation assay showed that miR396, miR166, miR6149-5P, novel133, or novel140 promoted *P. infestans* colonization, while miR394 inhibited colonization on *Nicotiana benthamiana* leaves. An artificial miRNA target (amiRNA) degradation experiment demonstrated that miR394 could target both potato gene (*PGSC0003DMG400034305*) and *P. infestans* genes. miR396 targets the multicystatin gene (*PGSC0003DMG400026899*) and miR6149-5p could shear the galactose oxidase F-box protein gene *CPR30* (*PGSC0003DMG400021641*). This study provides new information on the aspect of cross-kingdom immune regulation in potato-*P. infestans* interaction at the sRNAs regulation level.

## 1. Introduction

miRNA (microRNA) is eukaryotic endogenous, 20–24 nt non-coding single-stranded small RNA (sRNA), and the precursor is 70–80 nt single-stranded RNA with a hairpin ring structure. miRNAs are derived from single stranded RNA transcripts (MIR-genes), most of which are located in the intergenic region, and a few in the intron region [1]. miRNA plays a significant role in regulating plant growth and development, environmental stress, and disease resistance (against bacteria, fungi, oomycetes, etc.). Many studies have shown that potato miRNAs play an important role in the regulation of plant development, including the regulation of the conversion of potato stolon to tuber [2], the potato growth and development [3], and the anthocyanin synthesis [4]. Potato miRNAs were also involved in environment stresses by regulating target genes to adapt to abiotic stress, such as drought, salt, and cold [5,6].

Plants could regulate plant immunity using miRNAs to cope with the attack of pathogens [7,8]. miR6027, miR5300, miR476b, miR159a, and miR164a may play an important role in regulating tomato resistance against late blight disease [9]. Potato miRNAs, *stu*-miR482d-3p and *stu*-miR397-5P, regulate virus PVA resistance by targeting PR genes [10]. Potato miR160 regulates *P. infestans* infection and activates systemic acquired resistance (SAR) by regulating the salicylic acid (SA)-mediated resistance response [11]. In *Arabidopsis thaliana*, the interference of miR773 activity and concomitant up-regulation of its target gene *METHYLTRANSFERASE 2* (*MET2*) showed increased resistance to necrotrophic and hemibiotrophic fungal pathogens [12]. Decreased miR1916 expression in tomato could enhance the resistance against late blight and gray mold by increasing the expression of *isoodouin synthase* (*STR-2*), *UDP glycosyltransferase* (*UGTs*), and resistant protein genes *R1B-16*, *RPP13,* and *MYB12* [13]. miRNAs regulate the plant resistance immune response by regulating hormone signaling-related-genes. In *A. thaliana*, miR393 regulates plant PTI by regulating the auxin signaling pathway. Inoculation with non-pathogenic bacteria *Pst* DC3000 *hrcC* could rapidly induce the up-regulation of miR393, miR160, and miR167, and decrease the expression of the auxin response factor gene *ARF* [14]. The overexpression of miR172a and miR172b in tomato increases late blight resistance by inhibiting AP2/ERF transcription factor genes [15]. miRNAs could manipulate plant immunity by directly targeting resistance genes. nta-miR6019 and nta-miR6020 target the Toll and Interleukin-1 receptor-NB-LRR immune receptor tobacco *N* gene that confers resistance to the tobacco mosaic virus (TMV) [16]. Tomato miR482b directly targets the NBS-LRR gene, the lower expression of miR482b leads to stronger late blight resistance [17]. Expressions of *Sl*-miR482f and *Sl*-miR5300 in tomato were decreased during *Fusarium oxysporum* infection, and the expression of target NB-LRR genes were increased [18]. Meanwhile, plants can reduce miRNAs expression by targeting *R* genes to form a balance between plant growth and disease resistance. Cotton *ghr*-miR5272a limits excess immunity by inhibiting the expression of *MAPKK6* that encodes a protein to defend cotton against *Fusarium oxysporum* under normal growth conditions [19]. In tomato and other *Solanaceae* plants, the miR482/2118 family members target numerous NBS-LRR resistance protein genes. The pathogen-encoded RNA silencing suppressor can trigger the deletion of miR482/2118 in a nonspecific manner to release the inhibition of the NBS-LRR gene [20]. This balance enables resistant proteins to be expressed at a low level in the absence of pathogen infection and to be rapidly induced by pathogen infection [21].

Many studies have shown that miRNAs can be secreted into target cells through vesicles or binding proteins as cellular communication to regulate multiple target genes [22,23]. In addition to producing endogenous sRNAs to regulate their own virulence genes, pathogens can also produce sRNAs and migrate into host cells to target host immune-related genes during infection. In this process, the RNAi transboundary events of host-pathogen interaction open a new chapter in the movement and regulation of sRNAs between different species [24]. The transboundary behavior of sRNAs can be traced back to experiments in *Caenorhabditis elegans*, silent signals can cross cell boundaries and propagate between cells and tissues [25,26]. *Escherichia coli* bacteria expressing dsRNAs can interfere with the growth and development of nematode larvae that feed on them [25]. A recent study showed that plant miRNAs were transported into fungal pathogens through exosome vesicles [27]. *A. thaliana* secretory vesicles could carry sRNAs into *Botrytis cinerea* cells to silence important pathogenic genes [28].

A number of miRNAs were involved in immune regulation between tomato and *P. infestans* interaction [9]. Some tomato miRNAs have been reported to positively or negatively regulate tomato disease resistance against *P. infestans* [17]. It has been observed that many potato- and pathogen-derived sRNAs target resistance genes in the potato genome and *P. infestans* miR8788 was found to target a potato *StABH1* to promote potato late blight disease [29]. However, systemic information about miRNAs during potato-*P. infestans* interaction is still limited. In this study, the miRNA-Seq method was used to identify miRNAs during the interaction of potato and *P. infestans*. Both miRNAs and target genes were predicted in potato and *P. infestans*, quantitative real-time PCR was conducted to detect the expression of miRNAs, while artificial miRNA strategy was used to confirm the degradation between predicted miRNAs and their targets. This study provides useful information for the further exploration of the *P. infestans*-potato molecular interaction at a sRNAs cross-kingdom regulation level.

## 2. Results

### 2.1. sRNA Analysis during Potato-P. infestans Interaction

miRNAs from both hosts and pathogens manipulate plant immune responses and microbe pathogenicity by regulating the transcription and translation of target genes [24]. In order to explore the miRNAs and their regulatory target signal pathways in the process of *P. infestans*-potato interaction, potato leaves with a growth period of about 45 days were inoculated by the *P. infestans* strain HB0914-2 with a concentration of 100 spores/μL. RNA samples isolated from leaf samples after *P. infestans* inoculation for 24 h, 36 h, 48 h were used for small RNA sequencing. A total of 35, 537, 114 raw reads and 33, 746, 990 clean reads were obtained. After filtering out tRNA, rRNA, snoRNA, snRNA, etc., the remaining 19.8 million reads were mapped to the reference genome of potato DM1-3 and *P. infestans* T30-4, respectively (Figure 1A). It is clear that the most mapped reads were derived from potato (Figure 1A green bars); only a very small part of reads was mapped to the *P. infestans* genome (Figure 1A dark blue bars). The reads length distribution analysis showed that the most of sRNAs were mainly enriched between 22 and 24 nt, dominantly 24 nt sRNA (Figure 1B).

### 2.2. Prediction of miRNAs Expressed during Potato-P. infestans Interaction

The sRNA sequences mapped to the potato reference genome were further used to predict the miRNA by the software psRobot, mainly through screening whether the precursor formed a hairpin structure and whether the secondary structure was stable. A total of 171 miRNAs derived from potato were predicted (Appendix A). A sequence comparison of 171 predicted miRNAs with known plant miRNAs in miRbase showed that 43 miRNAs were annotated miRNAs (Table 1). The remaining 128 miRNAs were not annotated in the data base. They might be new potato miRNAs (Appendix A).

### 2.3. Most of Potato miRNAs Were Expressed in the Early Stage of Infection

A total of 171 potato miRNAs were identified during the *P. infestans* infection. According to the expression levels of these miRNAs at three time points (24 h, 36 h, 48 h), these miRNAs were divided into medium-high expression and low-expression miRNAs. The TPM (transcripts per million clean tags) values of 10 miRNAs were higher than 100, at least in one time point, and TPM values of 16 miRNAs were higher than 20 at least in one time point (Figure 2A,B). The expression level of the majority of miRNAs was higher at the 24 h time point (Figure 2C). Twenty-six high and medium expressed miRNAs were selected for further analysis. These miRNAs were clustered according to the expression levels at different time points. The results showed that the expression patterns of most miRNAs were the same. The expression was higher in the 24 h time point of infection and decreased across three time points (Figure 2D). The validation of the expression of five selected miRNAs using real-time PCR showed the same trends (Figure 3).

### 2.4. Targets Prediction of Potato-Derived miRNAs in Potato Genome

To investigate the potential targets of those predicted potato-derived miRNAs, the target genes of 171 potato miRNAs were predicted in the potato gene database. Totally, 171 miRNAs may target 1203 transcripts of 878 genes (Appendix A). Interestingly, half of those genes encode proteins directly related to resistance, including bacterial spot disease resistance protein 4, late blight disease resistance proteins RGA1, SH10 and CC-NBS-LRR resistance proteins, P450, BS2, Gpa2, I2C-5, R3a, RGA1/2/3/4, and RPI-BLB2, etc. Others, such as stu-miR482d-3p, stu-miR482e-3p, stu-miR6024-3p, and stu-miR6026-3p, were predicted to target CC-NBS-LRR resistance protein and Rpi-vnt1. Sixty-three genes may be involved in plant growth and development. Forty-seven were transcription factor genes. Forty-two predicted target genes were kinases, and 40 predicted target genes may participate in the anabolism of the cell wall and membrane (Figure 4). 

In order to explore the regulatory pathway of potato miRNA in regulating autoimmunity, GO (gene ontology) enrichment analysis was conducted. When biological processes were focused, a large number of genes were enriched in the innate immune response (GO: 0045087), defense response (GO: 0006952, GO: 0009814), cell death (GO: 0008219, GO: 0012501), and other pathways (Table 2). In the process of *P. infestans* infection, potato-derived miRNAs may not only affect the pathways that directly related to resistance and immunity, but also indirectly regulate immunity through manipulating transport and enzyme activity.

### 2.5. Targets Prediction of Potato-Derived miRNAs in P. infestans Genome 

Studies have shown that miRNAs can not only be transported to different cells through vesicles for target regulation, but even extend to cross-species regulation among different species [30]. However, it is not clear whether potato miRNA could regulate target genes of *P. infestans*. To investigate whether the potato miRNAs could target *P. infestans* genes, 171 potato-derived miRNAs were used to predict potential target 194 genes in the *P. infestans* genome; the result showed that 53 potato miRNAs might target *P. infestans* genes, including genes encoding transcription factors, kinase, ubiquitin proteins, and transmembrane proteins, but the function of most genes is unknown (Appendix A). Notably, three potato miRNAs (novel72, 133, 140) could only predict targets in the *P. infestans* genome (Table 3). These results implied that some potato miRNAs might specifically manipulate the expression of *P. infestans* genes.

### 2.6. miRNAs Regulate Plant Resistance against P. infestans

To investigate whether the potato-derived miRNAs were involved in regulating late blight resistance, 16 miRNAs were selected to test their function. The predicted target genes of seven miRNA (miR5300, miR482, mir482A-3p, mir482D-3p, mir482E-3p, mir6024-3p, and miR6027) are NBS-LRR resistance genes, so that the seven miRNAs tandem sequence was recombined into one vector as a positive control. All the primers for constructing miRNA expression vectors were shown in Appendix A. The mature miRNA expression skeleton was selected and combined with the mature miRNA sequence to generate the precursor sequence that could form a hairpin structure. The full-length sequence was recombined into the pK7LIC1.0 vector. Seven miRNAs tandem sequences (miR5300, miR482, miR482a-3p, miR482d-3p, miR482e-3p, miR6024-3p, and miR6027) were successively recombined into one vector and the other 16 selected miRNAs were separately recombined into the pK7LIC1.0 vector to produce plant expressing constructs. These vectors were transiently expressed in the leaves of *N. benthamiana* by agrobacterium-mediated infiltration, and an empty vector was used as a negative control. After 24 h of agro-infiltration, the leaves were detached and inoculated with the *P. infestans* strain 88069 with a concentration of 100 spores/μL. The disease lesion diameter was measured after 5–6 days. As expected, the transient expression of seven miRNA tandem sequences significantly promoted *P. infestans* colonization which was reflected by a bigger lesion diameter compared with that of EV control (Figure 5A). The results showed that only six (novel140, novel133, miR394, miR96, miR6149-5p, and miR166) out of the 16 independently expressed miRNAs altered plant resistance against *P. infestans*. Transient expression of novel140, novel133, miR166, miR396, and miR6149-5p promoted *P. infestans* colonization (Figure 5B–D,F,G). The transient expression of miR394 inhibited the colonization of *P. infestans* (Figure 5E). It is interesting that the predicted targets of novel133 and novel140 were *P. infestans* genes, but the transient expression of novel133 and novel140 could promote a *P. infestans* infection in *N. benthamiana* leaves. All the inoculation phenotypes were summarized in Appendix A. The above results demonstrated that some miRNAs identified in the potato-*P. infestans* interaction were involved in regulating the plant defense response.

### 2.7. Confirmation of the Degradation Ability of miRNA by amiRNAs Strategy

Potato-derived miR394, miR396, miR166, and miR6149-5p were predicted to target potato genes to regulate resistance against *P. infestans*. miR394, miR396, miR166, miR6149-5p, novel133, and novel140 were also predicted to target *P. infestans* genes. The transient expression of these miRNAs could affect the resistance of *N. benthamiana* against *P. infestans* (Figure 5). To prove whether these miRNAs could degrade the predicted target genes, artificial miRNAs (amiRNAs) strategy [31] was used in this study to test the miRNAs function for degrading target genes. By comparing the sequence of miRNA with predicted cDNA of target genes, the binding site was found, and the 24 nt sequence around the binding site was combined into a pMS4 vector in which the 24 nt sequence was fused with the green fluorescence gene *GFP* (Figure 6A). The expression vector of mature miRNA and the corresponding expression vector containing its artificial target fragment-GFP were co-expressed in *N. benthamiana* leaves; the fluorescence of GFP would be weakened or disappear if miRNA could degrade the artificial sequence designed according to the miRNA target gene. Otherwise, GFP is normally expressed. 

When miR394 was co-expressed with the artificial sequence of the potato target gene *PGSC0003DMG400034305*, the green fluorescence was weak compared with the control (Figure 6B). *PGSC0003DMG400034305* encodes an F-box protein that is associated with ubiquitination protein transferase activity, regulating the auxin-mediated signaling pathway and leaf vascular tissue morphogenesis. miR394 was also predicted to target multiple *P. infestans* genes (*PITG_08162, PITG_08164, PITG_08166, PITG_08167, PITG_08168, PITG_08169, PITG_08170, PITG_20983, PITG_20985,* and *PITG_20986*). These target genes are evolutionarily conserved in *P. infestans*, coding fibronectin type III domain proteins. The predicted *P. infestans* conserved sequence (here indicated by PITG) was inserted into the pMS4 vector fused with the green fluorescence gene *GFP*. The site co-expressing miR394 + pMS4-PITG showed a much weaker GFP signal compared to miR394 + EV (Figure 6C), implying that potato-derived miR394 has the potential to regulate *P. infestans* genes. According to GFP intensity, miR396 might degrade its potato target *PGSC0003DMG400026899* (Figure 6D), which encodes a multicystatin gene. miR6149-5p might degrade *PGSC0003DMG400021641*, which is reflected by the GFP intensity decrease compared with the control (Figure 6E). *PGSC0003DMG400021641* encodes a galactose oxidase, F-box protein CPR30. 

The above results demonstrated that the miRNAs identified in this study have the potential to degrade predicted target genes in- and cross-species and may participate in the regulation of late blight resistance.

## 3. Discussion

Late blight is one of the most important diseases affecting potato production. Aiming to look into the role of miRNAs and their immune regulation function in potato-*P. infestans* interaction, sRNA sequencing was conducted in this study. Totally, 171 potato miRNAs were found in sRNA-seq data, 43 of which were aligned to the potato miRNA database (Table 1). Meanwhile, 128 novel miRNAs were detected in this study (Appendix A). The number of known and novel miRNAs were varying widely compared to that in tomato responded to *P. infestans* infection [9]. Due to the different software and filter thresholds for miRNA, prediction is possible. *P. parasitica* and *P. infestans* produce diverse sRNAs to regulate endogenous genes [32,33] at stages of vegetative growth and during the host plant infection. In this study, a very small part of the reads (about 0.5 million) was mapped to the *P. infestans* genome (Figure 1A). Only three *P. infestans* miRNAs were predicted in this study (Appendix A). They show very low expression levels compared to other miRNAs. This is quite different from previous work [29]. We speculated that due to low *P. infestans* biomass and depth of sequencing. It is worth noting that, in research by Hu et al., Ago1 associated sRNAs were enriched by co-immunoprecipitation and deep sequenced during potato leaf infection [29].

We found that most of known miRNAs were highly expressed at 24 hpi, and decreased over time (Figure 2C,D). Among 128 novel detected miRNAs, novel135 and novel138, were highly expressed at 24 h, targeting histidine kinase and HD domain class transcription factor, respectively. However, most of the novel miRNAs were low expressed in our study, such as novel30, which is predicted to target the disease resistance protein genes *BS2*, *Gpa2*, and *PSH-RGH6*, which is likely to be important for regulating plant immunity. This implicated that the early infection stage is an important stage for plant immune system reprogramming by sRNA regulation. 

Most of known miRNAs and their function were consistent with previous work, such as miR156, miR159, miR171, miR5300, and miR6027 which were involved in plant immunity [9,34,35,36]. Some miRNAs could regulate *R* genes in plants directly, the miR482 family (stu-miR482a/d/e-3p and stu-miR482) mainly regulates NBS-LRR genes in eudicots, functioning as an essential component in plant disease resistance [37]; miR6024-3p regulates NLR genes to facilitate the disease caused by the necrotrophic pathogen [38]. In plants, high disease resistance often results in a reduction of growth. miRNA-mediated *R* gene turnover has been shown to be a protective mechanism to prevent autoimmunity in the absence of pathogens [39,40]. We noticed that about half of the predicted target genes of miRNAs identified in this study encode proteins that are resistance proteins, including bacterial spot disease resistance protein 4, late blight disease resistance proteins RGA1, SH10, and CC-NBS-LRR resistance proteins, P450, BS2, Gpa2, I2C-5, R3a, RGA1/2/3/4, and RPI-BLB2, etc. (Figure 4). It may demonstrate that potato miRNAs are expressed in normal conditions to balance *R* genes, inhibiting their expression without biotic stress. When the *P. infestans* signal is sensed by the sensing system, *R* genes will be released by the lower expression of miRNAs to conform a defense against pathogens. Besides regulating *R* genes, miRNAs may regulate plant development. *Brassica* miR1885 targets both an immune receptor TIR-NBS-LRR gene *BraTNL1* and a development-related gene *BraCP24* for negative regulation through distinct modes of action [41]. 

It has been shown that sRNA transboundary transportation and regulation play important roles during plant-microbe interaction. During *Verticillium dahliae* infecting cotton, cotton miR166 and miR159 expression were increased and transported into pathogen cells to silence pathogen genes *Clp-1* and *HCI-15* [42]. Pathogen sRNAs not only act on their own miRNA in the process of pathogen infection, but also can be transported to host species through vesicles to regulate the expression of other genes and further regulate the immunity [43]. Fungal botrytis sRNA *Bc*-siR37 suppresses plant defense genes by silencing several *Arabidopsis* target genes [44]. A large number of miRNAs were accumulated in *Cuscuta campestris* during the process of parasitizing *A. thaliana* and regulated multiple *A. thaliana* mRNAs [45]. In our study, sRNA sequencing and analysis implicated that potato or *P. infestans* miRNAs may manipulate their own genes and transboundary species genes. Fifty-three potato-derived miRNAs were predicted to target 194 *P. infestans* genes (Appendix A). Three potato miRNAs (novel72, 133, and 140) were only predicted targets in the *P. infestans* genome (Table 3) and two *P. infestans* miRNAs might target potato genes (Appendix A), implicated that the transboundary regulation of sRNAs in the potato-*P. infestans* interaction is possible, like other plant-microbe interactions [24]. It has been proven that Ago1-related *P. infestans* miRNA could target the potato *ABH1* gene and regulate potato resistance [29].

miR394 and its target gene *LEAF CURLING RESPONSIVENESS (LCR)* in *A. thaliana* responded to drought stress in an abscisic acid-dependent manner [46]. miR394 has been shown to be involved in plant development and inhibits leaf tilt in rice by targeting the leaf tilt gene (*LC4*) [47]. The stable transformation of miR394 in *Brassica napus* results in changes in fruit and seed development [48]. In this study, the transient expression of miRNA394 in *N. benthamiana* showed enhanced resistance against *P. infestans*. However, previous studies have shown that miR394 negatively regulates plant immunity by targeting *LCR* in *Arabidopsis* and tomato [49,50]. miR394 functions in negative modulation of *Fusarium oxysporum* resistancein garlic [51]. The result indicates that there are differences in disease resistance regulation for miR394 in different species, and this needs to be further determined through inoculation and the stable transformation of *N. benthamiana* and potato. The amiRNAs strategy confirmed that miRNA394 could degrade not only the target gene in potato, but also has potential to degrade the target genes of *P. infestans* (Figure 6B,C), suggesting that miR394 may regulate plant disease interaction through two different pathways during the process of the *P. infestans* infection to enhance potato resistance. Our result also shows that miR396 could target a multicystatin gene (*PGSC0003DMG400026899*) (Figure 6D). Previous work has shown that miR396 targets the growth regulatory factor gene (*GRF*), including *GRF*1/4/7/8/9 in *Arabidopsis*. Using synthetic simulated targets to reduce the expression of miR396 in *A. thaliana* leads to enhanced resistance to fungi [52], and miR396 was also involved in vernalization and floral organ development [53]. miR6149-5P may target a galactose oxidase gene (*PGSC0003DMG400021641*) (Figure 6E), which is involved in the biosynthesis of the cell wall, contributing to plant resistance. Gou et al. confirmed that miR6149-5P targets *CPR30* and is shown to be involved in plant resistance [54,55]. 

Advances of the cross-kingdom movement of miRNAs has shown great potential for modulating a plant’s disease response [56]. Although functions of most miRNAs identified in this study have not been verified, the miRNAs information provides new insights on the aspect of cross-kingdom immune regulation in potato-*P. infestans* interaction which will help us to develop a new approach to mitigate late blight disease. 

## 4. Materials and Methods

### 4.1. Plant Materials and Growth Conditions

*Nicotiana benthamiana* plants were grown in the growth room (16 h light/8 h dark) at 22 °C and 60% humidity. Four-week-old *N. benthamiana* plants were used for transient expression assays. Potato (*Solanum tuberosum* L.) plants were grown in the greenhouse under natural conditions. Potato leaves collected from about 45 days growth period plants were used for *P. infestans* inoculation and miRNA expression level tests.

### 4.2. P. infestans Inoculation and Infection Assay

The *P. infestans* isolates 88069 and HB09-14-2 were cultured and propagated on rye agar medium in petri dishes at 18–20 °C. The *P. infestans* mycelium-covered plates (grown about 14 d) were flooded with ddH_2_O and scraped to collect sporangia. The sporangia were filtered with 300 mesh nylon mesh, the spores were collected with 1200 mesh nylon mesh, and the concentration of the spores was adjusted by microscope inspection. Finally, the concentration of the spores for isolate HB09-14-2 was adjusted to 1 × 10^5^ mL^−1^ for potato leaf inoculation and the concentration of spores for isolate 88069 was adjusted to 1.1 × 10^5^ mL^−1^ for *N. benthamiana* leaf inoculation. The inoculation method was the same as Guo et al. [57].

For preparing potato sRNA-seq samples, a spore suspension of HB09-14-2 was evenly sprayed on the back of potato leaves, and samples were collected at the time points 24 h, 36 h, and 48 h. Three biological samples, each containing 10 leaves for each time point, were prepared. Samples were rapidly frozen in liquid nitrogen and stored at −80 °C for further use. 

For *N. benthamiana* leaf infection, ten-microliter droplets of 88069 zoospores were inoculated on the back of *N. benthamiana* leaves, and the leaves were placed on wet tissue paper in a sealed transparent box. The box was put in the dark at 20 °C for 24 h to promote *P. infestans* infection and then placed in a photoperiod environment of 16 h light/8 h dark for 5–6 d. The size of the disease lesion was measured according to the progress of the disease and compared with the control. 

### 4.3. sRNA-seq and Bioinformatics Analysis

All potato leaf samples were sent to Novogene (Novogene Bio Technology Co., Ltd., Nanjing, China) for RNA isolation and small RNA library construction and next-generation sequencing. The raw data of sRNA sequences was analyzed by the following procedure. Firstly, short sequences of other types of RNA are filtered out using blastn, such as tRNA, rRNA, snoRNA, and snRNA. The sequences of the filtered RNA were aligned to potato genome sequencing (*S. tuberosum* Group *Phureja* DM1-3 516 R44) [58] and the *P. infestans* T40-3 reference genome by bowtie [59]. The software psRobot [60] was used to predict miRNA. In order to predict the target genes of the predicted miRNAs, the script psrobot_tar was used to predict the target genes of the obtained miRNA according to the potato and *P. infestans* reference genome sequence. The predicted miRNA sequence was aligned to the predicted target transcript sequence, and the general base complementary sequence was the binding site.

### 4.4. Plasmid Constructs

To generate miRNA transient overexpression vectors, five pairs of primers (named P1, P2, P3, P4, and P5) were designed on the website of AMIRdesigner (http://lifenglab.hzau.edu.cn/Tools/index.php?tid=AMIRdesigner, accessed on 20 October 2020.). Among these, P1, P3, and P5 were common primers specific to each backbone sequence, while P2 and P4 were specific to the mature miRNA of candidates and backbone sequences. P1 and P2, P3, P4, and P5 were fused into two long chains respectively, then the obtained two strands DNA fragment was used as a template for PCR amplification using P1 and P5. The full-length fragment with a hairpin structure that could produce mature miRNA was obtained. The detailed reaction system and the reaction procedure were performed according to Yasir *et al.* [61]. The overexpression vector pK7LIC1.0, driven by the CaMV 35S promoter, was digested with *Sma* I restriction enzyme. The PCR products were recombined using T4 DNA ligase (Vazyme Biotech Co., Ltd., Nanjing, China) into pK7LIC1.0. 

To verify the degradation of predicted target genes by miRNA, a miRNA sensor construct was prepared. The miRNA sequence was aligned to the gene sequence, and the complementary region would be the potential binding site of miRNA. Specific primers (Appendix A) were designed to amplify the 24 nt target sequence which is complementary to the miRNA sequence. Two oligos bearing amiRCP binding sites were annealed together to generate a double-stranded DNA structure with “TCGA” and “CTAG” 5’ overhang at each end. The artificial target structure was then recombined with the pMS4 vector at *Xho* I and *Xba* I sites using T4 DNA ligase. The ligation reaction was transformed into *Agrobacterium tumefaciens* GV3101. 

The pK7LIC1.0 and pMS4 vectors were presented by Prof. Feng Li’s Lab, Huazhong Agricultural University. The vector maps were shown in Appendix A.

### 4.5. Agrobacterium-Mediated Transient Gene Expression Assays

A single clone of *A. tumefaciens* GV3101 containing expression vectors was selected and incubated in liquid YEB medium overnight at 28 °C. The cultured *Agrobacterium* suspension was centrifuged at 4000 rpm for 10 min and resuspended with MMA (10 mM MES, 10 mM MgCl_2_ and 200 μM acetosyringone). The concentration of the suspension was measured by spectrophotometer OD_600_. The final concentration of OD_600_ was 0.1 for the transient expression and *P. infestans* infection assay. For the degradation of predicted target genes by miRNA assay, the miRNA vector final concentration of OD_600_ was 0.1 and the final concentration for the target vector OD_600_ was 0.05. The agro-infiltration method was the same as Zhou et al. [62].

### 4.6. RNA Extraction and Quantitative Real-Time RT-PCR Analysis

Total RNA was extracted from the leaves using a TransZol kit (Trans, Beijing, China), and the cDNA was synthesized using the miRNA 1st Strand cDNA Synthesis Kit by stem-loop (Vazyme Biotech Co., Ltd., Nanjing, China) according to the manufacturer’s instruction. Quantitative real-time PCR (qPCR) was carried out using the miRNA Universal SYBR qPCR Master Mix (Vazyme Biotech Co., Ltd., Nanjing, China). *U6* was used as a reference control gene for the miRNA qRT-PCR analysis. Nine potato leaves were sampled in each *P. infestans* inoculation time point, three leaves were pooled into one biological replicate, resulting in three biological replicates. Gene expression levels were analyzed by the 2^−△△CT^ method utilizing *U6* as the reference gene.

### 4.7. Statistical Analyses 

All data and statistical analysis were carried out using one-way ANOVA and pairwise or multiple comparisons with the Graphpad Prism 8.0 software (GraphPad Prism software Inc. San Diego, CA, USA, www.graphpad.com). All values and error bars presented are means ± SD or SEM of three or more experimental replicates. 

## Figures and Tables

**Figure 1 plants-12-00461-f001:**
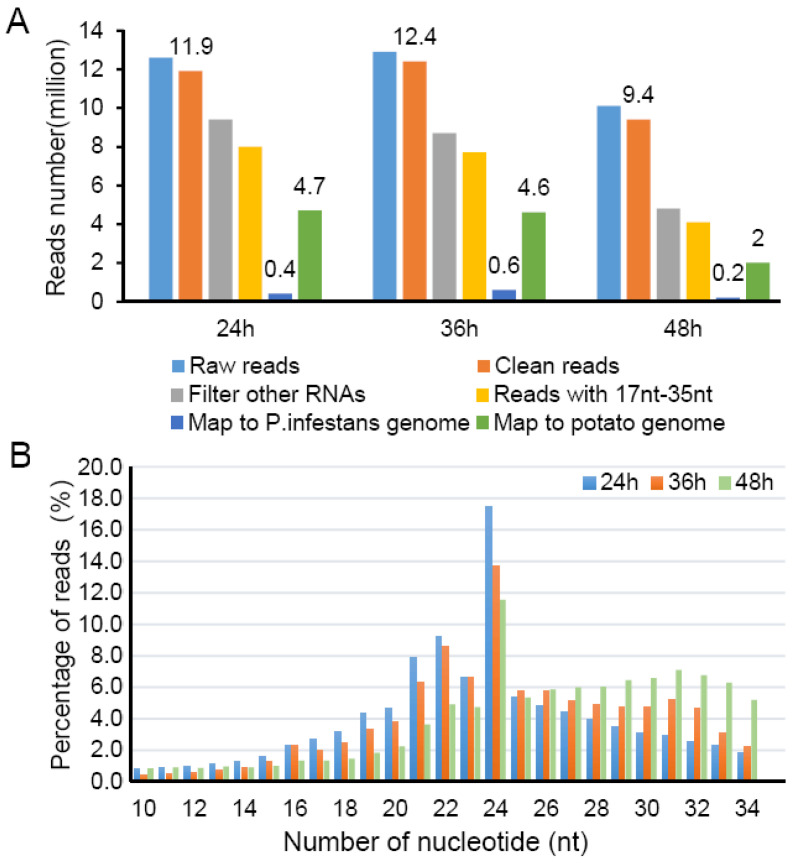
Summary of basic information of the sRNA-seq data. (**A**) sRNA reads distribution at different time points after inoculation with *P. infestans*. (**B**) Length distribution of clean sRNA reads. The x-axis shows the sRNA sequence sizes, from 10 to 34 nt; the y-axis shows the percentage of reads for every given size. Bars with different colors show different time points.

**Figure 2 plants-12-00461-f002:**
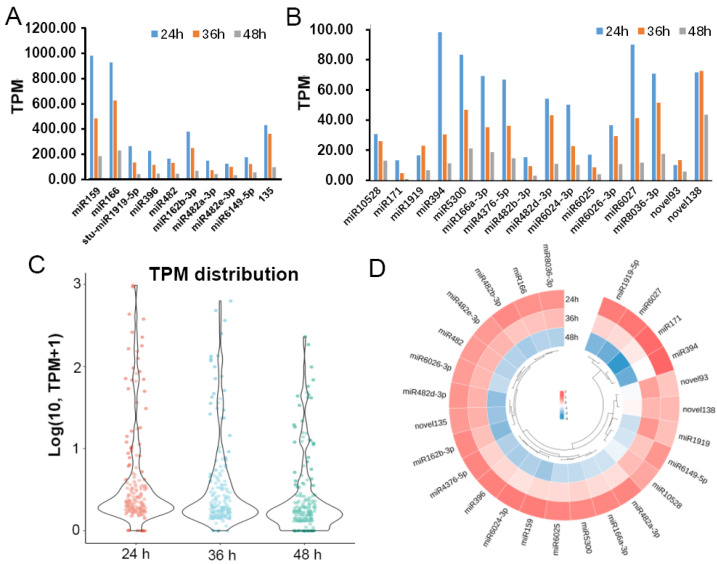
The expression profile of potato miRNAs in three time points after *P. infestans* inoculation. (**A**) The expression of 10 high-level-expressed miRNAs at 24 h/36 h/48 h. (**B**) The expression of 16 medium-level expressed miRNAs at 24 h/36 h/48 h. (**C**) Boxplot of the log transformed TPM (transcripts per million clean tags) expression values of 171 miRNAs. (**D**) Hierarchical clustering of the 26 differentially expressed miRNAs.

**Figure 3 plants-12-00461-f003:**
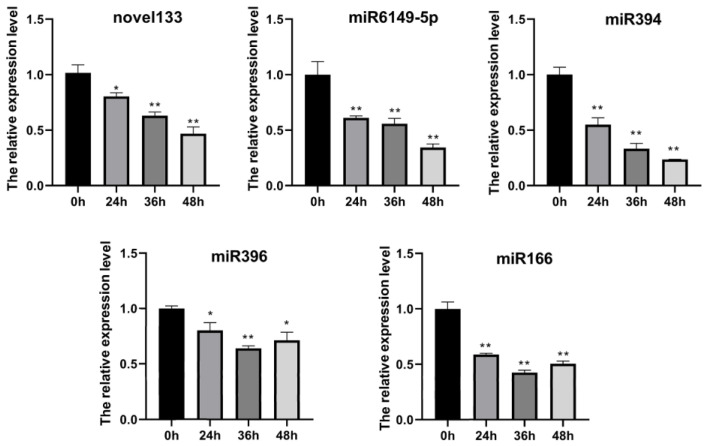
Real-time quantitative PCR validation of 5 miRNAs during *P. infestans* infection three time courses. miRNA expression level was analyzed by the 2^−△△CT^ method utilizing *U6* as the reference gene. One-way ANOVA, *, *p* < 0.05, **, *p* < 0.01. Three potato leaves were pooled into one biological replicate at each *P. infestans* inoculation time point. Error bars represent mean ± SD of three biological replicates.

**Figure 4 plants-12-00461-f004:**
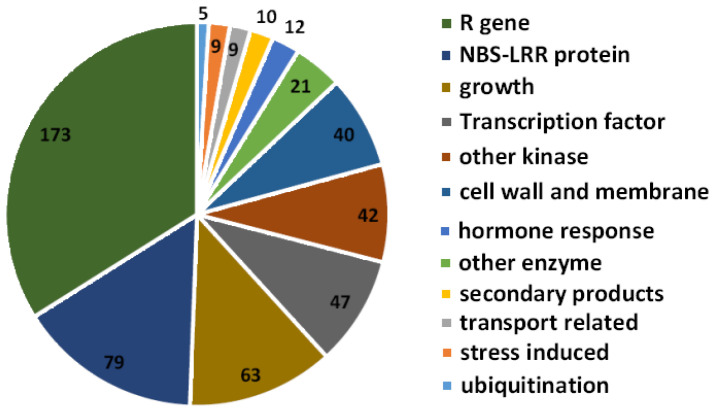
The classification of predicted target genes with known function in potato.

**Figure 5 plants-12-00461-f005:**
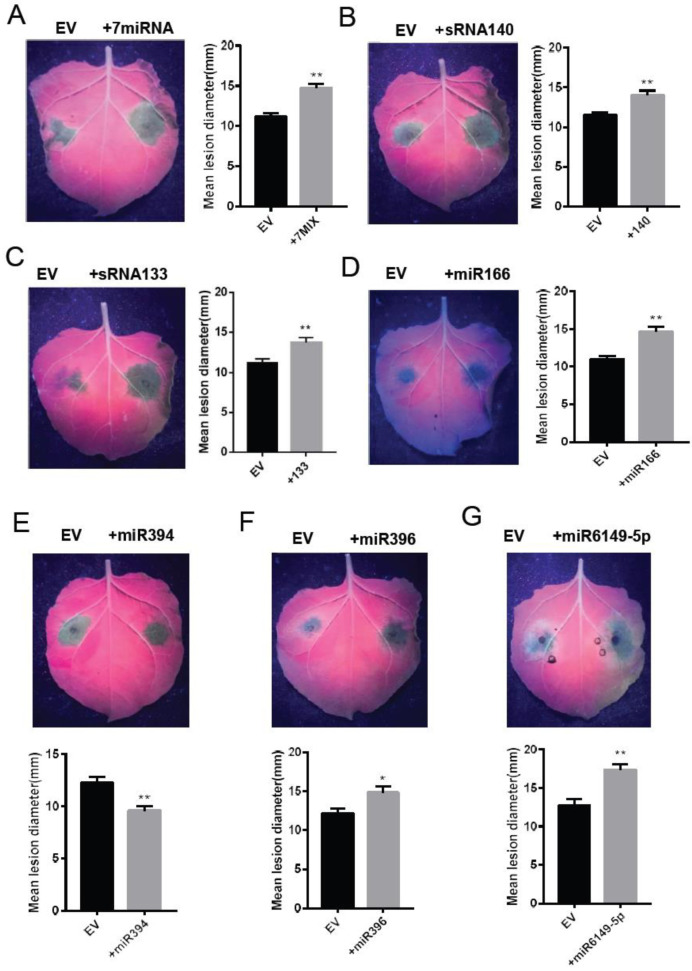
Transient expressing miRNAs to verify their function in regulation of late blight resistance. (**A**) 7miRNA means tandem expression vector containing seven miRNA (miR5300/miR482/miR482a-3p/miR482d-3p/miR482e-3p/miR6024-3p and miR6027). (**B**–**G**) novel140, novel133, miR166, miR394, miR396, or miR6149-5p was agro-infiltrated in *N. benthamiana* half of leaf, and empty vector (EV) was agro-infiltrated in another half of leaf. The leaf photos were taken under UV-light. Bar graphs on the right of leaf showed disease lesion diameter. miRNAs and EV were agro-infiltrated in *N. benthamiana* leaves for 24 h before *P. infestans* strain 88069 inoculation. Disease lesion was measured 5–6 d after *P. infestans* inoculation (one-way ANOVA, **, *p* < 0.01, *, *p* < 0.05, three biological replicates with 20 leaves from 6–7 plants at least for each replicate). Error bars represent mean ± SEM.

**Figure 6 plants-12-00461-f006:**
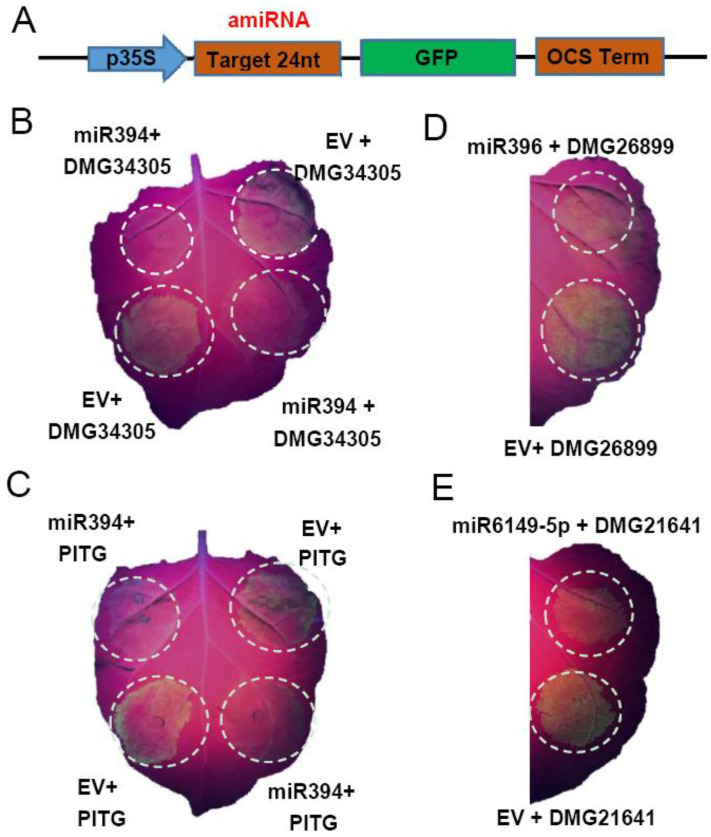
miRNAs suppress the expression of artificial sequences of predicted target genes. (**A**) Schematic expression structure of the amiRNA. Artificial sequence of predicted target gene was fused with green fluorescence *GFP*. When this structure was co-expressed with miRNA in *N. benthamiana*, if miRNA could cleave the artificial target sequence, the *GFP* will not be expressed normally and GFP fluorescence signal would be weak or disappear. (**B**) Sites co-expression of miR394 with its artificial target sequence of *PGSC0003DMG400034305* showed weak GFP fluorescence signal compared to co-expression of EV with the artificial target sequence. (**C**) Sites co-expression of miR394 with *P. infestans* artificial target sequence (PITG) showed weak GFP fluorescence signal compared to co-expression of EV with the target sequence. (**D**) Site co-expression of miR396 and its artificial target sequence of *PGSC0003DMG400026899* showed that miRNA396 could suppress the expression of GFP. (**E**) Site co-expression of miR6149-5P and its artificial target sequence of *PGSC0003DMG400021641* showed weak GFP fluorescence signal compared to co-expression of EV with the target sequence. Three leaves of each miRNA assay show similar performance.

**Table 1 plants-12-00461-t001:** Predicted potato miRNAs that are already annotated in miRbase during potato-*P. infestans* interaction.

No.	miRNA	24 h	36 h	48 h	No.	miRNA	24 h	36 h	48 h
1	miR10528	29.89932	25.37756	12.92606	23	miR482a-3p	148.6614	74.19915	40.70108
2	miR156	0.918694	0.805637	0.320481	24	miR482b-3p	15.11669	9.506514	3.311636
3	miR159	980.3301	482.8181	185.3448	25	miR482d-3p	52.5326	41.97368	10.89635
4	miR162b-3p	378.669	248.7001	67.72831	26	miR482e-3p	124.1072	99.8184	33.00954
5	miR162b-5p	8.351764	4.511566	1.709232	27	miR530	1.4198	0.241691	0
6	miR166	927.2129	624.3685	230.5326	28	miR5300	80.59453	45.43792	20.72443
7	miR166a-3p	67.06467	34.32013	18.37424	29	miR6024-3p	48.69079	22.23558	10.25539
8	miR166b	6.764929	3.06142	1.06827	30	miR6025	16.70353	8.620314	4.166252
9	miR167b-3p	2.338494	1.852965	0.427308	31	miR6026-3p	35.66203	28.60011	10.78953
10	miR167d-3p	3.090153	3.464238	1.06827	32	miR6027	87.02538	40.12071	11.64414
11	miR171	13.19579	4.994948	1.281924	33	miR6149-5p	175.9717	121.9734	56.51147
12	miR171b-3p	2.422012	1.691837	0.640962	34	miR7122-3p	1.252765	0.322255	0.320481
13	miR1919	16.28594	22.47727	6.836927	35	miR8011a-3p	0.835176	0.483382	0.106827
14	miR319-3p	2.338494	8.056368	11.75097	36	miR8016	7.767141	1.611274	1.602405
15	miR394	94.95956	29.64743	11.21683	37	miR8033-5p	1.586835	0.644509	0.320481
16	miR396	226.4163	115.6089	47.21753	38	miR8036-3p	68.56798	50.03005	17.30597
17	miR397b-3p	0.918694	0.563946	0.320481	39	miR8036-5p	0.668141	0.161127	0.320481
18	miR398	2.422012	2.578038	0.427308	40	miR8044-3p	1.002212	0.8862	0.106827
19	miR398b-3p	3.006635	5.317203	0.961443	41	miR8051-5p	0.584623	0.402818	0.213654
20	miR4376-5p	64.72617	35.28689	14.52847	42	miR1919-5p	264.5004	134.3802	41.12839
21	miR477b-5p	1.002212	0.725073	0.534135	43	miR8039	1.586835	2.336347	1.495578
22	miR482	164.6133	132.0439	44.86733					

Note: miRNAs expression level shown by TPM (transcripts per million clean tags) value.

**Table 2 plants-12-00461-t002:** GO analysis of predicted target genes of potato-derived miRNAs.

	GO ID	Description	NO.	Bg	*p*.Value
CC	GO:0005669	Transcription factor TFIID complex	2	7	3.30 × 10^−1^
	GO:0030127	COPII vesicle coat	2	14	3.83 × 10^−2^
	GO:0005886	Plasma membrane	96	2741	2.40 × 10^−4^
MF	GO:0003955	NAD(P)H dehydrogenase (quinone) activity	2	3	3.64 × 10^−2^
	GO:0015152	Glucose-6-phosphate transmembrane transporter activity	2	3	3.64 × 10^−2^
	GO:0010542	Nitrate efflux transmembrane transporter activity	3	6	7.48 × 10^−3^
	GO:0015562	Efflux transmembrane transporter activity	5	36	4.71 × 10^−2^
	GO:0004872	Receptor activity	36	192	2.27 × 10^−18^
BP	GO:0048263	Determination of dorsal identity	2	2	3.91 × 10^−2^
	GO:0010072	Primary shoot apical meristem specification	5	20	1.05 × 10^−2^
	GO:0009814	Defense response, incompatible interaction	20	171	6.20 × 10^−6^
	GO:0009626	Plant-type hypersensitive response	63	173	1.08 × 10^−50^
	GO:0045087	Innate immune response	71	412	2.68 × 10^−33^
	GO:0007154	Cell communication	80	1757	6.63 × 10^−4^
	GO:0012501	Programmed cell death	171	410	1.13 × 10^−156^
	GO:0008219	Cell death	173	475	8.70 × 10^−147^
	GO:0006952	Defense response	240	1529	1.09 × 10^−116^

Note: CC means cellular component; MF means molecular function; BP means biological process; Bg means the number of background genes in this pathway.

**Table 3 plants-12-00461-t003:** Predicted *P. infestans* genes only targeted by three potato-derived miRNAs.

Potato miRNA	*P. infestans* Gene	Annotation
novel 72	*PITG_15260*	Conserved hypothetical protein
	*PITG_15263*	Transmembrane protein
novel 133	*PITG_05949*	Conserved hypothetical protein
	*PITG_07309*	Flagellar radial spoke protein
	*PITG_09123*	Conserved hypothetical protein
	*PITG_12774*	Conserved hypothetical protein
novel 140	*PITG_08205*	Methionyl-trna formyltransferase

## Data Availability

The original data sets described in the study are included in the article/Appendix A. Further inquiries can be addressed to the corresponding author.

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
