# Peer review of "Identification of miRNAs Involving Potato-Phytophthora infestans Interaction"

_plants, 2023, doi:10.3390/plants12030461_

Round 1

Reviewer 1 Report

Congratulations on completing this study. You made attempt to identify miRNAs in potato-Phytophthora infestans interaction, vis RNAseq which is novel input to the scientific field, there is a lack of research perspective in interaction study. Such as given in fig.4 only selected mi RNA was analyzed, even though their interaction, should be shown with more clarity, specifically in defense response genes. Figure 5. - There is missing data for the validation of key defense gene response upon miRNAs treatment, at least the interesting miRNAs treatment(sRNA140,sRNA133, Mi394) effect on plant immunity genes must be studied by RTqPCR.

Figure 6- improved figures should be replaced as the GFP difference among control and treated is not visible. Author claims in results must be in specific words Line- 276, Above results demonstrated that miRNAs identified in this study have potential to degrade predicted target genes in cross-species, and may participate in the regulation of late blight resistance. The discussion needs improvement in the last part which can show how your findings show cross-kingdom interaction, also schematic diagram addition would be great to explain reader-friendly.

There should not be mistakes such as font and bold at lines 389-395.

Best of luck.

Author Response

Congratulations on completing this study. You made attempt to identify miRNAs in potato-Phytophthora infestans interaction, vis RNAseq which is novel input to the scientific field, there is a lack of research perspective in interaction study. Such as given in fig.4 only selected miRNA was analyzed, even though their interaction, should be shown with more clarity, specifically in defense response genes.

Response: Thank you very much for your positive comments and suggestion. In this study we aimed to identify miRNA involved in manipulation of late blight resistance. Because we have not verified their certain functions yet, we did not express too much perspective.

Figure 4 shows function categories of all potato target genes predicted by potato-derived miRNAs. Examples of specific miRNAs and their target R genes were added in text content. And all R genes and defense response genes target by miRNA were shown in Table S2.

Figure 5. - There is missing data for the validation of key defense gene response upon miRNAs treatment, at least the interesting miRNAs treatment (sRNA140,sRNA133, Mi394) effect on plant immunity genes must be studied by RT-qPCR.

Response: We agree with your wonderful advice. In figure 5, we attempted to show that some of identified miRNA may involve in resistance regulation by transient expression of miRNAs. We think it is not reliable to test key defense gene response upon miRNAs in transient expression system. We are now try to producing stable transgenic potato plants expressing specific miRNAs for verify their function and to validation the key defense gene response as you suggested in coming work.

Figure 6- improved figures should be replaced as the GFP difference among control and treated is not visible.

Response: Sorry for that. We have selected the best images. Because GFP expression is not too stronger, the GFP signal difference between control and treated is not very clear. But we can see the stronger GFP signal in EV + amiRNA-GFP site rather than in miRNA + amiRNA-GFP site.

“Author claims in results must be in specific words Line- 276, Above results demonstrated that miRNAs identified in this study have potential to degrade predicted target genes in cross-species, and may participate in the regulation of late blight resistance.” The discussion needs improvement in the last part which can show how your findings show cross-kingdom interaction, also schematic diagram addition would be great to explain reader-friendly.

Response: Thank you for your suggestion. We observed that miR394 could degrade both the artificial target sequence of potato gene (PGSC0003DMG400034305) and P. infestans gene, so we think that “miRNAs identified in this study have potential to degrade predicted target genes in cross-species”.

Line- 276: this sentence has be changed to “Above results demonstrated that miRNAs identified in this study have potential to degrade predicted target genes”. In discussion we avoided to emphasize that.

Legend of schematic diagram in Figure 6 was improved to make it clear for reader. 

There should not be mistakes such as font and bold at lines 389-395.

Response: the format mistake has been corrected in the text.

Congratulations on completing this study. You made attempt to identify miRNAs in potato-Phytophthora infestans interaction, vis RNAseq which is novel input to the scientific field, there is a lack of research perspective in interaction study. Such as given in fig.4 only selected miRNA was analyzed, even though their interaction, should be shown with more clarity, specifically in defense response genes.

Response: Thank you very much for your positive comments and suggestion. In this study we aimed to identify miRNA involved in manipulation of late blight resistance. Because we have not verified their certain functions yet, we did not express too much perspective.

Figure 4 shows function categories of all potato target genes predicted by potato-derived miRNAs. Examples of specific miRNAs and their target R genes were added in text content. And all R genes and defense response genes target by miRNA were shown in Table S2.

Figure 5. - There is missing data for the validation of key defense gene response upon miRNAs treatment, at least the interesting miRNAs treatment (sRNA140,sRNA133, Mi394) effect on plant immunity genes must be studied by RT-qPCR.

Response: We agree with your wonderful advice. In figure 5, we attempted to show that some of identified miRNA may involve in resistance regulation by transient expression of miRNAs. We think it is not reliable to test key defense gene response upon miRNAs in transient expression system. We are now try to producing stable transgenic potato plants expressing specific miRNAs for verify their function and to validation the key defense gene response as you suggested in coming work.

Figure 6- improved figures should be replaced as the GFP difference among control and treated is not visible.

Response: Sorry for that. We have selected the best images. Because GFP expression is not too stronger, the GFP signal difference between control and treated is not very clear. But we can see the stronger GFP signal in EV + amiRNA-GFP site rather than in miRNA + amiRNA-GFP site.

“Author claims in results must be in specific words Line- 276, Above results demonstrated that miRNAs identified in this study have potential to degrade predicted target genes in cross-species, and may participate in the regulation of late blight resistance.” The discussion needs improvement in the last part which can show how your findings show cross-kingdom interaction, also schematic diagram addition would be great to explain reader-friendly.

Response: Thank you for your suggestion. We observed that miR394 could degrade both the artificial target sequence of potato gene (PGSC0003DMG400034305) and P. infestans gene, so we think that “miRNAs identified in this study have potential to degrade predicted target genes in cross-species”.

Line- 276: this sentence has be changed to “Above results demonstrated that miRNAs identified in this study have potential to degrade predicted target genes”. In discussion we avoided to emphasize that.

Legend of schematic diagram in Figure 6 was improved to make it clear for reader. 

There should not be mistakes such as font and bold at lines 389-395.

Response: the format mistake has been corrected in the text.

Reviewer 2 Report

It is well known that while most sRNAs function endogenously, some can travel across organismal boundaries between hosts and microbes and silence genes in trans in interacting organisms, a mechanism called ‘‘cross-kingdom RNAi.’’ During the co-evolutionary arms race between fungi and plants, some fungi developed a novel virulence mechanism, sending sRNAs as effector molecules into plant cells to silence plant immunity genes, whereas plants also transport sRNs, mainly using extracellular vesicles, into the pathogens to suppress virulence-related genes or use these sRNAs to modulate their own defense response after pathogen attack.

The authors of this manuscript identified miRNAs in the context of a potato-Phytophtora infestans interaction/infection. They identified (only) potato miRNAs, some of which were already annotated in the miRNA database as well as novel potato miRNAs and they predicted target genes in potato and P. infestans for these miRNAs and tested the functions of a few of them.

Although the methods and the research design described by the authors are appropriate and the conclusions concerning the potato miRNAs drawn are supported by the results I do have some concerns:

The interaction of potato and P. infestans on the miRNA level is already extensively examined but the authors lack to discuss some of this relevant literature to an appropriate degree, e. g. Hu et al, 2022, New Phytologist, 233, 443-457.

However, the more serious concerns relate to the fact that the authors found only a few P. infestans miRNAs and those with a very low expression level. However, it is known from other publications that P. infestans in the context of potato infection may well produce a variety of miRNAs, even with a significantly higher expression level than that described by the authors in this manuscript. The most prominent of these miRNAs is miR8788, which was, however, not found by the authors of the present manuscript. This casts doubt on their results. In my opinion, it is therefore necessary for the authors to prove the existence of low expressed P. infestans miRNAs described in their manuscript as well as  the presence of miR8788 not found by them, e.g. by Northern blot in the material produced by them.

Author Response

It is well known that while most sRNAs function endogenously, some can travel across organismal boundaries between hosts and microbes and silence genes in trans in interacting organisms, a mechanism called ‘‘cross-kingdom RNAi.’’ During the co-evolutionary arms race between fungi and plants, some fungi developed a novel virulence mechanism, sending sRNAs as effector molecules into plant cells to silence plant immunity genes, whereas plants also transport sRNs, mainly using extracellular vesicles, into the pathogens to suppress virulence-related genes or use these sRNAs to modulate their own defense response after pathogen attack.

The authors of this manuscript identified miRNAs in the context of a potato-Phytophtora infestans interaction/infection. They identified (only) potato miRNAs, some of which were already annotated in the miRNA database as well as novel potato miRNAs and they predicted target genes in potato and P. infestans for these miRNAs and tested the functions of a few of them.

Although the methods and the research design described by the authors are appropriate and the conclusions concerning the potato miRNAs drawn are supported by the results I do have some concerns:

The interaction of potato and P. infestans on the miRNA level is already extensively examined but the authors lack to discuss some of this relevant literature to an appropriate degree, e. g. Hu et al, 2022, New Phytologist, 233, 443-457.

However, the more serious concerns relate to the fact that the authors found only a few P. infestans miRNAs and those with a very low expression level.

However, it is known from other publications that P. infestans in the context of potato infection may well produce a variety of miRNAs, even with a significantly higher expression level than that described by the authors in this manuscript. The most prominent of these miRNAs is miR8788, which was, however, not found by the authors of the present manuscript. This casts doubt on their results. In my opinion, it is therefore necessary for the authors to prove the existence of low expressed P. infestans miRNAs described in their manuscript as well as  the presence of miR8788 not found by them, e.g. by Northern blot in the material produced by them.

Response: Thank you very much for your critical comments.

In this study we attempted to identify miRNA from both potato and P. infestans during their interaction. But prominent of miRNAs identified were from potato. There are, may be, several reasons, such as low biomass of P. infestans in our samples, sRNA sequencing and predict methods. One thing we noticed is that the amount of sRNA reads mapped to P. infestans genome is too small, and among of them only 3 sRNAs were predicted to form hairpin structure and with the stable secondary structure. We speculate that the samples and method we used might not suitable to identified P. infestans miRNA.

7238 P. infestans sRNAs were identified by Fahlgren et al. (2013). In this study pure mycelium of P. infestans was used to isolate total RNA.

In the literature (Hu et al, 2022), a specific transgenic P. infestans strain was used in this study. This stain harbouring P. infestans Ago1-GFP (pHAM34:PiAgo1-GFP) and PiAgo1-GFP was used to co-immunoprecipitate small RNAs from total RNA isolated from mycelium of P. infestans or from total RNA isolated from P. infestans inoculated potato leaves. In such way, P. infestans Ago1-associated sRNAs were highly enriched, which make it easy to identify more miRNA.

Hu et al, (2022) stated that “Among the 7238 sRNAs from P. infestans, the only previously confirmed Pi-miRNA (miR8788) (Fahlgren et al., 2013) was detected in the PiAgo1-GFP infected leaf sample, but not in the pHAM34:GFP infected sample.” Interestingly, in Fig. 2d, even É£32P labelled RNA probe was used, it is hard to detect miR8788-5p in P. infestans infected potato leaves with strains 88069 and pHAM34: GFP compared to pure mycelium samples.  Authors also expressed that “Throughout our work, Pi-miR8788-3p has for some inexplicable reason not been possible to detect by Northern blot, although it is present in the pHAM34:PiAgo1-GFP co-IP datasets. Equal levels of miR8788-5p and miR8788-3p were observed in agroinfiltrated N. benthamiana leaves (Fig. 3d), highlighting the importance of analysis in a correct biological context (e.g. in a suitable host and in presence of the pathogen) to differentiate the active from the inactive miRNA species.”

So we think that different biological context would be take into account for detecting low level miRNA expression.

Before we used Dig-labeled probes to detect miRNAs in total RNA, due to the low detection sensitivity, it is not easy to do that. A ɣ32P labelled RNA probe is necessary in our future work.

Fahlgren et al. Phytophthora Have Distinct Endogenous Small RNA Populations That Include Short Interfering and microRNAs. PLoS ONE, 2013, 8, e77181.

Hu et al. Phytophthora infestans Ago1-associated miRNA promotes potato late blight disease. New Phytol. 2022, 233, 443-457.

Reviewer 3 Report

I would be very grateful to get the authors explain why among 53 potato miRNA they have selected 16 miRNA to test their functions?

Also, I recommend to do some corrections as follow:

Line 19-20. Three potato miRNAs (novel72/133/140) were predicted, and two potentially target

Line 213. Table S5 instead Table S4

Line 299. Table S1 instead Table S2

Author Response

I would be very grateful to get the authors explain why among 53 potato miRNA they have selected 16 miRNA to test their functions?

Response: Thank you very much for your kind comments!

According to the expression level (high and medium) and the function of their predicted targets in defense response, 16 miRNA were selected to test their function.

Also, I recommend to do some corrections as follow:

Thanks a lot for point out our mistake! 

Line 19-20. Three potato miRNAs (novel72/133/140) were predicted, and two potentially target

Line 213. Table S5 instead Table S4

Line 299. Table S1 instead Table S2

Response: We have corrected them.

Round 2

Reviewer 2 Report

I´m not very satisfied with the author´s response. Basically they only added two sentences to their discussion as a reaction to my critical remarks. I still think that the authors should do some additional experimental work (e. g. working with a É£32P labelled RNA probe to detect P. indestans miRNAs and not Dig-labeled probes as they claim themselves) before the manuscript should be published.

Author Response

Thank you very much for your critical comments.

Sorry for our inappropriate response!

Frankly, we wanted very much to do the experimental work to test P. infestans miRNAs expression as you suggested. But due to the safety control reason, we cannot easily order radiative 32P at that moment. Because only 3 P. infestans miRNAs with low expression level were predicted and we cannot confirm that. We think that the relevant result was not reliable as you mentioned. In this manuscript, most results were based on the potato-derived miRNAs. So we deleted the relevant result (table 2 and table 3) in main text and put them in table S1. And we only mentioned that in discussion part.

Hoping for your understanding.